# Data Portraits: Recording Foundation Model Training Data

**Marc Marone**    **Benjamin Van Durme**
Johns Hopkins University
{mmarone1,vandurme}@jhu.edu

## Abstract

Foundation models are trained on increasingly immense and opaque datasets. Even while these models are now key in AI system building, it can be difficult to answer the straightforward question: has the model already encountered a given example during training? We therefore propose a widespread adoption of *Data Portraits*: artifacts that record training data and allow for downstream inspection. First we outline the properties of such an artifact and discuss how existing solutions can be used to increase transparency. We then propose and implement a solution based on data sketching, stressing fast and space efficient querying. Using our tools, we document a popular language modeling corpus (The Pile) and a recently released code modeling dataset (The Stack). We show that our solution enables answering questions about test set leakage and model plagiarism. Our tool is lightweight and fast, costing only $3\%$ of the dataset size in overhead. We release a live interface of our tools at dataportraits.org and call on dataset and model creators to release Data Portraits as a complement to current documentation practices.

## 1 Introduction

Modern AI is driven by large models trained on large datasets, displaying emergent capabilities as they scale [5]. Despite the foundational nature of these models, it is a natural question to ask of some example, *"Was this seen during training?"* Understanding the contents of large, opaque, web-scraped datasets is important to assessing downstream model behavior. Web datasets can contain leaked test sets, harmful text, or low quality information; these impact downstream models [12, 26, 15, 8]. Documentation artifacts for datasets and models have been proposed and adopted by the community [14, 27, among others]. However, less work has dealt with tooling to support large dataset inspection.

We suggest that existing practices around foundation models can be improved through the widespread adoption of *Data Portraits*: artifacts that record training data and allow for efficient inspection. The critical property of a data portrait is *membership inference*: whether an example was part of a data collection. In discussing these tools we consider three perspectives: content creators (owners), scientists, and content consumers. Creators wish to know whether a dataset contains their content (e.g. copywritten code). Scientists may seek to understand model behavior by assessing test set leakage or knowledge memorization. Content consumers (including downstream applications) may want to know if a model is plagiarizing or citing existing resources.

Our contributions are two-fold. First we call for documentation artifacts based on membership inference, discussing the properties and tradeoffs around existing solutions. We then introduce our own implementation of such an artifact and use it to document a widely used large language model (LLM) corpus: the Pile [13]. We also document a recent code language modeling corpus: the Stack [22]. Our artifact uses data sketching (compressed or approximate views of data, [6]) to enable millisecond latency and minimal compute requirements, using only $\sim 3\%$ of the original dataset size. We release our tools and a live demo at dataportraits.org.

37th Conference on Neural Information Processing Systems (NeurIPS 2023) Track on Datasets and Benchmarks.

Table 1: Properties of various membership testing tools. Our proposed implementation, the strided Bloom filter, is probabilistic because it uses hash-based matching, but avoids lossless redistribution of data. ~ indicates that a tool might support a property. See Table 2 and Section 3 for further discussion about performance.

| Method | Avoids Redistribution | Local | Index | Fuzzy | Probabilistic |
|---|---|---|---|---|---|
| grep | ✗ | ✓ | ✓ | ✗ | ✗ |
| Full Text Index | ✗ | ~ | ✓ | ~ | ✗ |
| Suffix Array | ✗ | ✓ | ✓ | ✗ | ✗ |
| Strided Bloom Filter | ✓ | ✓ | ✗ | ✗ | ✓ |

## 2 Background and Related Work

### 2.1 Documentation Artifacts

Much recent work has called for additional artifacts documenting datasets and models [2, 27, 14, among others]. Gebru et al. [14] argues that creators should release a Datasheet artifact documenting the *"motivation, composition, collection process"* for a dataset. Mitchell et al. [27] earlier proposed a related artifact called Model Cards for trained models. These suggestions have been taken up within the AI community: new models are released with these artifacts (e.g. Zhang et al. [37] provide a Datasheet and Model Card for their models intending to be an open source parallel to OpenAI's closed models) and many resources hosted by Huggingface now include documentation artifacts.[1]

We view a Data Portrait as a complementary type of documentation artifact - one that answers the membership question. There already exist examples of what we would describe as a portrait. Dodge et al. [12] studied multiple aspects of the C4 corpus and released a searchable indexed version.[2] Contemporaneously with this work, Piktus et al. [30] released a search tool for the ROOTS corpus [23]. These tools are increasingly important as web-text corpora continue to outscale the resources of many academic groups.

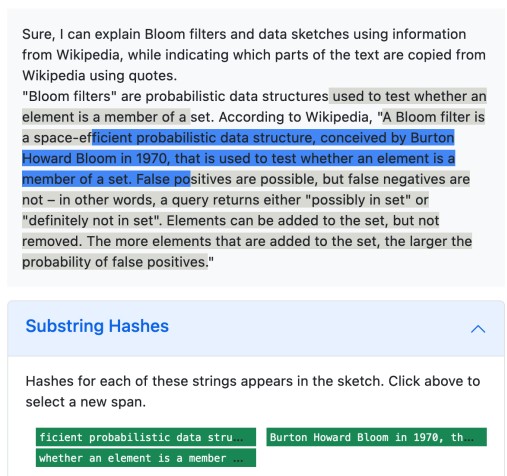

Figure 1: Output from ChatGPT [29] when asked to explain Bloom filters using text from Wikipedia. The highlighted string is the longest overlapping string with the Pile. Other overlapping spans are grey. See our demo: dataportraits.org

**Datasheets** Gebru et al. [14] acknowledge that the contents of a complete datasheet will vary depending on research circumstances.

> *... datasheets will necessarily vary depending on factors such as the domain or existing organizational infrastructure and workflows ... [such as] academic researchers publicly releasing datasets ... [or] product teams creating internal datasets for training proprietary models.* – Gebru et al. 14

For all of these scenarios, even internal proprietary uses, it is beneficial to have a computationally verifiable form of documentation.

**Model Cards** The Model Cards form seeks to document training data with several sections, including the one duplicated below:

---

[1] https://twitter.com/mmitchell_ai/status/1548358023382347777
[2] https://c4-search.apps.allenai.org/

*Training Data. May not be possible to provide in practice. When possible, this section should mirror Evaluation Data. If such detail is not possible, minimal allowable information should be provided here, such as details of the distribution over various factors in the training datasets.* – Mitchell et al. [27]

We strongly prefer that datasets be open and transparent. However in cases where this is impossible, certain implementations of Data Portraits can still allow for (approximate) membership testing on the dataset used to build a specific model. In Section 3.1, we describe how some implementations can be used without leaking or redistributing data.

## 2.2 Large Models and Web Data

Other related work analyzes properties of models trained on web-scale corpora. Carlini et al. [9] showed that GPT-2 memorizes and can leak sensitive or private information from its training corpus. Further work [10] examined memorization and showed that it increased with scale. The initial GPT-3 [8] construction suffered from a bug that meant certain testsets were not filtered from their very large training corpus (though they later carefully analyzed the effects of this bug). Conventions and legal precedents around large language models that may memorize and output portions of the training data are still being developed. Kocetkov et al. [22] gather a set of permissively licensed open source code repositories. Luccioni and Viviano [26] analyze a typical source of web data, the Common Crawl, studying "content that can be generally seen as inappropriate for a language model to generate".

It is not always clear what was used for training existing models. Even though open source models such as OPT [37] provide documentation artifacts, their Datasheet simply notes that the training corpus is a filtered *"union of the following datasets..."* and they do not host a final accessible artifact.[3]

## 2.3 Data Structures

Data sketching, storing compressed or approximate views of large datasets, has long been used to enable efficient analysis of large data [6]. GPT-3 and other LLMs ([25, 23] among others) use the MinHash sketch of Broder [6] to deduplicate their training sets. However, they do not release this sketch for downstream use as a documentation tool. Other data sketches, such as Bloom filters [4] and related structures, have a long history of use as efficient storage for NLP tasks. Talbot and Osborne [34] use Bloom filters to construct language models while others used similar structures to count features [16, 36]. More recent work analyzes the use of a Bloom filter to prevent verbatim copying of common n-grams from training sets [19].

Recent work on large datasets has investigated deduplication and memorization analysis. Lee et al. [24] develop an optimized suffix array implementation that can scale to very large datasets. Other work has released traditional full text indexes for datasets they study [30, 12]. These structures can be used as a membership testing tool but are primarily meant for additional use cases and thus may have higher computational demands (see Table 2 and Section 3).

# 3 Data Portraits

We outline desirable properties of a dataset documentation tool built around membership inference, recognizing that some may induce tradeoffs.

1. **Fast:** An artifact should support low latency programmatic access.
2. **Lightweight:** An artifact should not be much more expensive than storing the original data.
3. **Avoids Redistribution:** Some data cannot be redistributed for legal or proprietary reasons. A tool should avoid propagating harmful or illegal content when possible.
4. **Local:** A local artifact requires no costly external services. A local tool also provides privacy when querying for PII (personally identifiable information), rather than transmitting PII to a third party.
5. **Indexing:** A tool should indicate the context in which a match is found.

---

[3]https://github.com/facebookresearch/metaseq/issues/20

6. **Match Flexibility:** Fuzzy matching can be useful for similarity searches. Exact matches might be more useful for cases involving PII or memorization.

**Other Implementations**   Examining these, it is clear that there may be many solutions to creating membership inference tools on large datasets. Table 1 has an overview of properties related to data modeling (discussed below) while Table 2 compares space usage between other published tools.

The easiest version of a Data Portrait is to simply store the data on disk. grep or similar tools can be used to query the dataset. This is a typical approach for membership inference; Carlini et al. [9] mention they ask the GPT authors for grep results. Other implementations might include a full-text search using commercial software or a conventional database [30, 12]. Recent work on memorization and data duplication has explored efficient data-structures for deduplicating large text datasets [24].

While these techniques enable tools that we would consider viable Data Portraits, each has certain tradeoffs. Simply storing the corpus on disk and grep-ing is slow and only allows exact or regex matches. This also does not address the governance issue - some data cannot be redistributed. For example, there are restrictions on distributing tweets [35]. However, since grep iterates over the corpus, indexing comes naturally.

A full-text search engine typically relies on an efficient indexing and lookup method and can be very fast. However, this takes additional space or might depend on non-local services. It also does not address the redistribution issue. A full-text search (e.g. BM-25 in Lucene) can support fuzzy or exact match searches, though these may add additional overhead and efficiency concerns.

Structures such as a suffix array are suitable for fast, local use on a single machine but have very high space requirements. Lee et al. [24] use a suffix array to deduplicate C4 [33] but note that *"this algorithm still requires that the dataset itself fits in memory"* and suggest using a machine with >600GB of RAM. These structures allow for indexing into the dataset but do not support fuzzy matching. A local suffix array redistributes data; if exposed as a service, enumerating and extending queries could extract full documents.

Given these tradeoffs, we use a *probabilistic* approach for membership inference: a data sketch. A sketch provides a compressed and approximate view of data [4, 6, 7]. We use hash-based matching, addressing the redistribution issue, but giving up the ability to index or perform fuzzy matching. Our solution is minimal in that it supports membership inference and nothing more – yielding a small and fast structure. Other tools might support additional features or index the original documents. See Table 2 for a size comparison with other tools, such as the ROOTS Search Tool BM25 index of Piktus et al. [30]

Table 2: Ratio of structure to dataset size for related tools that could be used for membership testing. Tool & Data Sizes in TB ($10^{12}$ bytes).

| Structure | Dataset | Tool | Data | Ratio |
|---|---|---|---|---|
| grep | (any) | - | - | 1.0 |
| RST (BM25) | ROOTS | 2.78 | 1.54 | 1.80 |
| Suffix Array | C4 | 1.65 | 0.86 | 1.92 |
| Ours | Pile | 0.03 | 0.89 | **0.03** |

on the ROOTS corpus [23]; and the Suffix Array of Lee et al. [24] on C4 [33]. Most tools are built for *indexing*, which causes the structure size to be at least as large as the data. *Our minimal implementation uses only $\sim 3\%$ the space of the original corpus with millisecond query latency.*

## 3.1   Bloom Filter Sketching

We base our sketch implementation of a portrait on Bloom filters [4]. These are a well-known solution for space efficient approximate membership testing, using only a fixed number of bits to record elements of a set. For example, our sketch of the Pile uses only $\sim 14$ bits per data element. Bloom filters are similar to hash tables, but they store only the hash of a data element rather than storing the element itself. This is accomplished by setting bits in an array indexed by hash functions. See Broder and Mitzenmacher [7] for further details and derivations of hash collision rates. Bloom filters can output false positives but never false negatives. The false positive rate can be tuned to optimize the final size of the structure.

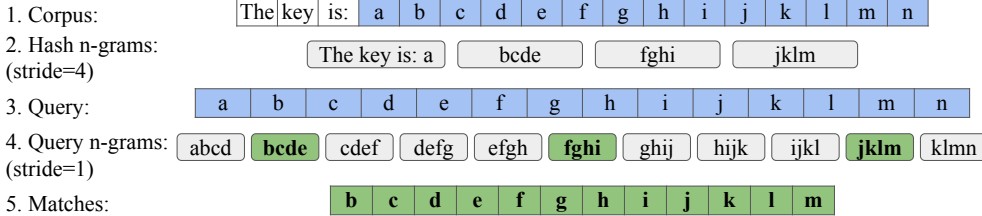

Figure 2: The matching process for a Bloom filter sketch with n-gram width $4$. The blue tokens *abcdefghijklmn* are a string of interest. Checking subsequences indicates several matches (green).

## 3.2 Bloom Filter Construction

Since our use case is membership testing over subsequences of longer documents, we can modify the input to our Bloom filter. Instead of storing all n-grams we store only tiled (or strided) n-grams from a source corpus. When building our Bloom filter data sketch, we hash n-grams with stride $n$ (i.e. non overlapping, see Figure 2 rows 1 and 2). This implicitly defines an offset or alignment where an n-gram starts (row 2). Each resulting strided n-gram is hashed and stored in the Bloom filter (row 2). At query-time, we extract n-grams of size $n$ but stride $1$ from the input query (Figure 2 rows 3 & 4). Each of these is checked against the Bloom filter and matches are recorded (row 4, green).

This reduces space requirements but carries the risk of missing some string of interest if it is split by a tile boundary in the corpus. This boundary problem can be alleviated by querying for sufficiently long strings and setting an appropriate size of $n$. Observe that if a string of interest is at least of length $2n - 1$ it will necessarily contain at least one tile of size $n$, producing a match. However, there may be hanging tokens at the start and end of a string. See Supplementary Appendix B for an example.

**Chaining**   We can further *chain* a set of matching query n-grams into longer sequences. If matches occur $n$ indices apart, they can be joined as a single *inferred* string (final row, Figure 2). This does not guarantee that the n-grams occurred in that particular order, but this is unlikely with long chains of long n-grams. Permutation attacks could produce bags of features that would harm other retrieval or indexing tools. See Appendix C for further discussion of adversarial permutations. Examining chains of matches at a document level alleviates the risk of false positives: a single n-gram match might be due to a false positive, but the probability of a chain of such matches decreases exponentially with the chain length.

**Redistribution**   Previously we noted that other tools for inspecting data redistribute the content, leading to legal and privacy concerns. Since Bloom filters distribute only hashes, they provide some obfuscation. Bianchi et al. [3] term this "Better Than Nothing" privacy and discuss information hiding bounds on Bloom filters with various parameters. They note that the protection offered by one-way hashing is vulnerable when *"the universe set is easily enumerable"*. In our case, this is either unicode sequences of length $n$ or sequences $V^n$ for tokens in a vocabulary set $|V|$. Storing strided n-grams provides additional information hiding without changing the size of the universe. Rather, it becomes harder to guess another member element given an existing match. If we stored every n-gram, an attacker could reconstruct documents by simply guessing single token extensions to an existing member. Since we store strided n-grams, an attacker would need to guess a sequence of length $n$ to find the next element in a document. Note that in principle, proprietary LLM providers (e.g ChatGPT [29]) could release a Bloom filter based Data Portrait *without revealing their exact training data – only the hashes of some substrings.*

## 4   Case Studies

We construct several Data Portraits based on Bloom filters and demonstrate case studies. With the chosen parameters, these sketches are most suitable for checking document level queries. A live interactive interface is available at dataportraits.org.

## 4.1 Constructing Portraits

We use an optimized ingestion pipeline that requires only a single modest machine. Our pipeline supports both token and character-based sketches. We apply simple whitespace normalization that enables exact string matching on code snippets at different indentation levels. We use Redis[4] to store the resulting hashes.

Further examples in this paper use a sketch built on character n-grams of length 50. This n-gram width was chosen for two reasons. First, it is in-line with related work on dataset filtering and de-contamination. GPT-2 uses 8-grams to analyze test set leakage and GPT-3 uses between 8 and 13-grams for similar analysis based on properties of the test set [32, 8]. We find that an 8-gram corresponds to approximately 52 characters of text in the Pile. Secondly, to further validate this choice of parameter, we analyze a large subset of the Pile and find that out of all extracted 50-grams, only $\sim 1.6\%$ of extracted n-grams are duplicated elsewhere in the subset. This value is $7.5\%$ for 25-grams and $79\%$ for 10-grams. In effect, a 50 character span is already a good document fingerprint. We also found that this width tends to match meaningful spans of text rather than spurious matches of common short spans. See Section 5.1 for further examples.

**The Pile**   We construct a strided Bloom filter on The Pile [13] which is built on 825 GiB of documents. Certain subdatasets are re-weighted and the final dataset is distributed as 1254 GiB of text, after decompression. Our text pipeline takes around 12 hours on a machine with 40 cores and 140 GB of RAM. The token level version takes approximately 24 hours on the same machine, including the time to decompress and tokenize the Pile. The structure takes only 27 GB on disk and has a false positive rate of $1 \times 10^{-3}$ (see Section 5.1 for further analysis of false positives). We note that these compute resources are fairly modest and our pipeline is stream based - the only requirement is sufficient RAM to hold the final structure. See Table 2 for an outline of the space usage of other structures that could be used for membership inference. With hash-based matching, we use $\sim 14$ bits per data element (50 UTF-8 characters).

**The Stack**   The Stack is a collection of permissively licensed code data obtained by downloading GitHub repositories [22]. We use the subset of The Stack used to train StarCoder, a 15.5B parameter large language model for code [25]. Specifically we process the data after test-set decontamination and preprocessing ($\sim$800 GB of code). The Stack Portrait uses approximately the same amount of compute as the Pile Portrait, producing a 26GB documentation artifact.

## 4.2 WMT Overlap

Recent work on large language models (LLMs) has found that they learn language translation implicitly through supervision present in web-scale training sets [8, 37]. A reasonable reaction is to be suspicious that models have memorized target text from the training corpus. We scan WMT (Workshop on Machine Translation) test sets available through SacreBLEU [31], reconstructing documents by concatenating lines. Since line concatenation does not restore paragraph boundaries present, this is only a lower bound on overlap. We analyze non-English references since source documents are typically published English news articles.[5]

Given this collection of documents, we search for overlap using our Pile Bloom filter-based Data Portrait. We chain individual matching n-grams as described in Section 3.2. Recording the max length of a found chain gives us a measure of *approximate longest overlapping subsequence*. Results are shown in Figure 3 for the 10 longest matches with a more detailed table in Appendix D. Most of the test documents with high overlap are older test sets, where test data may have been publicly available for some time (e.g. on GitHub or rehosted by other projects). One exception is the WMT 2020 English Inuktitut [1] test set (bold). A full document from this test set appears in the Pile.[6] The exact training data is not known for many LLMs: we cannot conclude whether translation targets are memorized by a specific model, but our Pile-based sketch shows that some test data has leaked.

---

[4] https://redis.io/
[5] Some WMT sets include reverse-created text [17]. Only recent LLM work [18] has analyzed this effect.
[6] Original Article

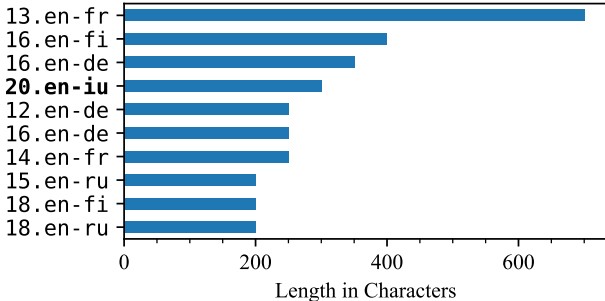

Figure 3: Approximate longest overlapping subsequences between individual documents in WMT test sets and the Pile. Y-axis is year and language.

## 4.3 Overlap Detection

Following Dodge et al. [12, Table 2] we quantify the degree of overlap for selected test sets and the Pile. We do not seek to exactly replicate their table as there are several fundamental differences — they use full text exact match against C4 [33] and we use a measure of expected 50-gram overlap against the Pile.

Additionally we only select datasets with minimal normalization and preprocessing since these are likely to appear in web-scrapes. Specifically we use target text from XSum [28], TIFU [20], and AMR2.0 [21, LDC2017T10]. XSum and TIFU are summarization datasets created from news articles and reddit respectively. Labels consist of brief summaries of a larger document. AMR2.0 (Abstract Meaning Representation) is a treebank resource that can be used for graph-to-text generation. Inputs consist of semantic graphs and labels are a target sentence with the same semantics. Some target text is translated to English specifically for this resource and should not appear naturally in a web-scrape absent leakage.

Table 3: Overlap statistics. E.O. is the Expected Overlap metric. Time is in seconds, measuring the total query time for that dataset.

| Dataset | %E.O. | Instances | Time |
|---|---|---|---|
| XSum | 40.12% | 11.3K | 3.61 |
| TIFU-short | 4.12% | 79.7K | 1.28 |
| TIFU-long | 3.86% | 42.1K | 10.73 |
| AMR2.0 (LDC) | 8.12% | 1.4K | 0.45 |
| AMR2.0 (Plain) | 8.06% | 1.4K | 0.43 |
| Sum | - | 136.0K | 16.5 |

We use an *Expected Overlap* metric that compares the expected number of n-gram matches in a document (had the complete document appeared in the Pile) to the observed longest match. Given a strided Bloom filter sketch with width $w$, *Expected Overlap* for a corpus $T$ is: $\frac{\sum_{d \in T} \max(chains_d)}{\sum_{d \in T} E(length(d), w)}$ where $d$ are documents in a test set $T$. $max(chains_d)$ is the max chain length i.e. the approximate longest overlapping subsequence between a document and the sketched dataset. $E()$ is the expected number of matches had $d$ appeared entirely in the corpus, accounting for boundary issues (see Appendix E). Many other metrics are possible using our sketch (e.g. total match count).

Results are in Table 3. Target summaries from XSum have the highest incidence of overlap with the Pile. This is unsurprising since the dataset is constructed from easily scraped BBC articles [28]. TIFU (short and long) are constructed from summaries of reddit posts. Many of these summaries are too short to guarantee a match in the strided filter we use here for illustration, so these overlap statistics are a lower bound. For AMR2.0, we search for both raw target text as it appears in LDC (Linguistic Data Consortium) distributed files and detokenized plaintext. We find slightly higher overlap for the raw text. This suggests that some LDC files are publicly exposed and scraped into LLM corpora (LDC data licenses typically forbid redistribution).[7] Creating this table is extremely fast using our sketch, taking on the order of 0.1ms per instance (136K total). While the methods are not directly comparable, available full text search tools take on the order of seconds per query when accessed on the web.

---

[7]We manually found instances of LDC data on GitHub.

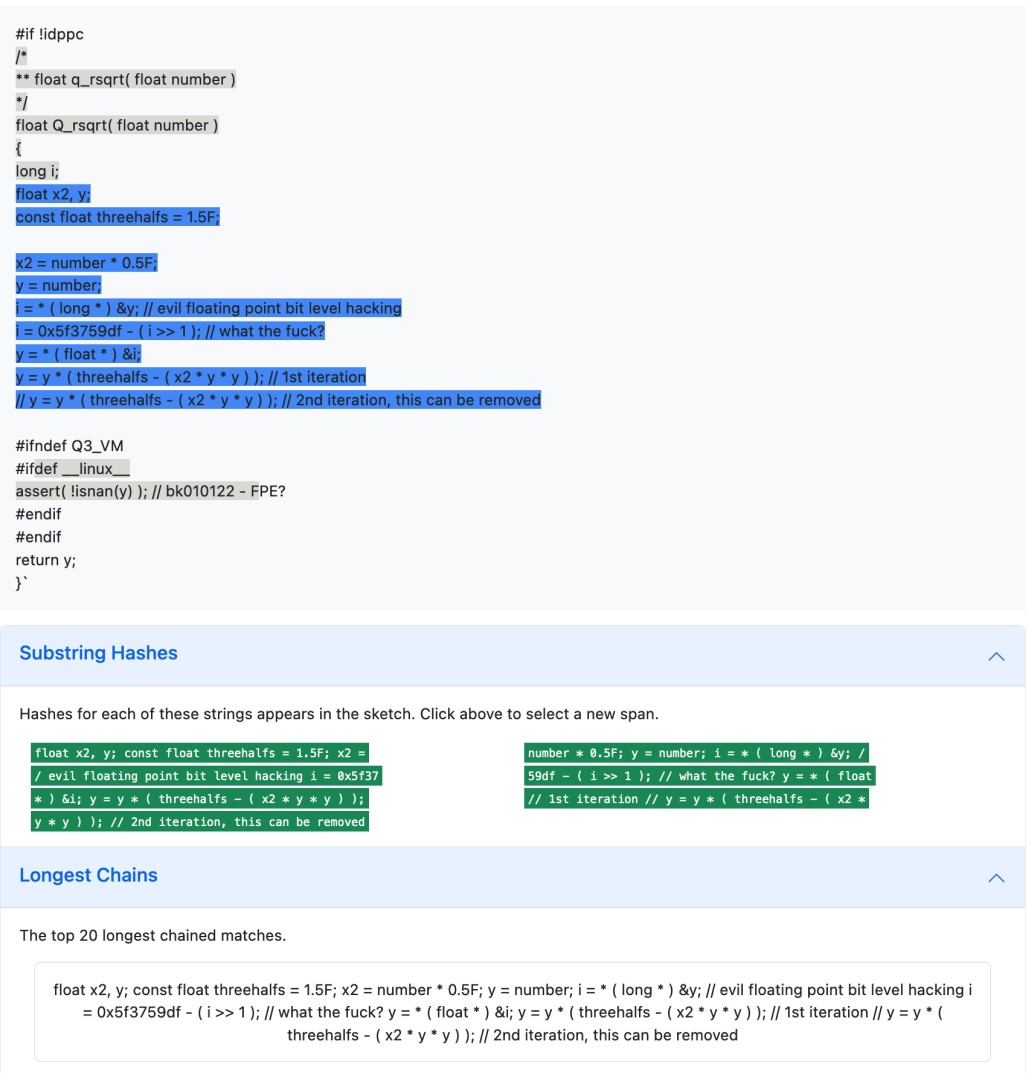

Figure 4: Text for the Fast Inverse Square Root algorithm from the Quake III source code. The blue highlighted span is the longest match, component spans are in green below. Some of the grey spans are preprocessor directives that also appear in the Pile.

## 4.4 Code Generation

Code generation or text-to-code models are another application of LLMs. GitHub Copilot[8] is a tool built on the OpenAI Codex model [11]. Some users have found that Copilot will copy famous snippets of code and add a new license.[9] GitHub has studied plagiarism and has taken steps to avoid copying from existing repos [38]. However, conventions and precedents around language models trained on open source code are rapidly developing. Content creators (i.e. programmers who have published code on GitHub) may wish to check if their code has been included as part of an LLM training corpus. Similarly, a downstream user might want to check if the outputs of a tool like Copilot substantially overlap a collection of data (e.g. for licensing compliance purposes). Figure 4 shows an example of our full front-end on a famous snippet of code. See the demo at pile.dataportraits.org

---

[8]https://github.com/features/copilot
[9]https://twitter.com/mitsuhiko/status/1410886329924194309

to interact with this example. The highlighted text suggests that the passage appears in the Pile. Specifically, hashes for the contiguous 50-grams are present in the pile (green, below). The *Longest Chains* section lists the longest sub-strings composed of these n-grams. Since hashing is very fast, users can interact with this interface and see updates at typing speed.

We build a Portrait documenting The Stack in addition to our Pile Portrait. This documentation artifact is available through our web interface (stack.dataportraits.org) and additionally powers a plagiarism checking service exposed to live users. This artifact exactly documents the training data used in StarCoder, a 15.5B parameter code generation model [25]. The StarCoder model has been used in a variety of open source applications and tools including a Copilot-like autocomplete plugin for VSCode.[10] This plugin implements a plagiarism check by calling our Portrait service. Spans that overlap the model's training set are highlighted in the user's editor. Our Portrait service acts as a lightweight first-pass check for model plagiarism and additional tools from Li et al. [25] can be used to further assess attribution. Our sketch-based Data Portrait is efficient – this service has active users and runs on a single cloud VM with 32GB of RAM. Most queries take only $\sim 10$s of milliseconds.

## 5 Analysis

To further validate our approach, we return to the fundamental stakeholder question: "is this text in a corpus?" This is a binary classification task: is query document $d$ in dataset $C_{pile}$? We design a simple experiment to test how well our method method works as a classifier and compare it to a full text search engine corresponding to the Roots Search Tool of [30].

Table 4: $F_1$ and latency on our classification task. Both methods can correctly classify Pile text, but ours (**BF**) is very fast.

| Method | $F_1\uparrow$ | Sec/Doc$\downarrow$ |
|---|---|---|
| **BF** | 1.0 | 0.015 |
| **FTS**$_{200}$ | 1.0 | 11.28 |
| **FTS**$_{50}$ | 0.57 | 1.81 |

**Data** We construct a small test set of text sampled from the Pile and text that is certainly not in the Pile. Without prior access to a membership testing tool, collecting text not included in an LLM corpus can be difficult. We harvest paragraphs from New York Times articles published between August 7-18 2023. Since the Pile was created in 2020, this text is certainly not in the Pile and any overlap should be spurious (e.g. long proper nouns or repeated quotations). This set consists of 5.5k `wc -w` words and is balanced with respect to document length and class label.

**Classifiers** The strided Bloom filter approach produces a list of n-grams and a boolean membership value. As described in Section 3.2, we can obtain an approximate longest common substring ($lcs$) between the query and corpus. Our classification rule is simply: $\frac{length(lcs)}{length(query)} > 0.9$ where 0.9 is a boundary threshold. We construct a similar classifier using a full text Lucene index on the Pile, where we take the common substring between the query and top result.[11] We find both classifiers can achieve perfect precision and recall on this synthetic task. However, the full text search classifier (FTS in Table 4) is much slower than the Bloom filter based classifier (BF).[12] The retrieval engine's response time is heavily dependent on query length, thus we experiment with only the first $N$ characters of the document: FTS$_N$. At 200 characters of context, the FTS classifier achieves perfect recall and precision though it takes $\sim 700x$ more time than the Bloom filter. Queries are much faster with 50 characters of context, but precision and recall suffer. We again emphasize that full indexes provide far more information than a Bloom filter; however, a minimal Bloom filter is sufficient for this classification task.

### 5.1 False Positives

Our method is probabilistic but the previous sections show that it is still useful for understanding datasets. Here we further analyze false positives. Using the dataset of novel text constructed for the previous section, we record all n-grams for which the Bloom filter returns `True`. To obtain ground

---

[10]https://github.com/huggingface/huggingface-vscode

[11]This search instance did not directly support exact match phrase queries.

[12]Latency measurements are averaged over 5 runs to account for network timing.

truth labels for $ngram \in C_{pile}$, we query the Pile Lucene index and record whether any of the top 25 documents contain that n-gram.

We first find that there are *no chained matches when querying our system with novel New York Times text.* That is, only isolated 50-grams match, rather than a sequence of multiple matching n-grams. Using the ground truth labels from the Pile search, we next find a false positive rate of $7 \times 10^{-4}$ (slightly less than the intended rate of $1 \times 10^{-3}$). The NYTimes text also contains some true positive matches. For example, ``National Counterintelligence and Security Center, '' is a 50-character string that coincidentally appears in both our novel corpus and the Pile. We found that a resolution of 50 characters is sufficient for document fingerprinting, but this parameter could also be increased to reduce matches of long common strings. We further note that the false positive rate (FPR) is simply a parameter that can be directly set (at construction time) at the cost of space. Decreasing the FPR by an order of magnitude (to $1 \times 10^{-4}$) would increase storage size to about 36 GB and increasing the FPR to $1 \times 10^{-2}$ would result in an 18 GB structure.

## 6    Impact and Limitations

We hope that this work will encourage members of the field to adopt membership testing tools as part of a complementary suite of dataset and model documentation artifacts. Many in the field have taken issue with massive opaque datasets. Our web interface and tools are meant to facilitate simple, easy, transparency. We hope that one broader impact of our demo will be enabling both NLP specialists and lay-people in assessing large language models. For example, a non-technical content creator could search for their material in a documented dataset.

Our work is limited in that we focus on studying only a few datasets. We hope our tooling will enable the study of other datasets in the future. We stress a lightweight and efficient hash-based approach to membership testing, but this comes with certain trade-offs such as false positives and the inability to retrieve the context surrounding a match. We also note that while we intend to better document existing datsets, these tools could be used to modify generated outputs. That is, an open plagiarism detection tool could be used to avoid plagiarism detection.

## 7    Conclusions and Future Work

Foundation models come with concerns such as plagiarism, test-set and personal data leakage, data contamination, and data provenance. We have argued that existing documentation practices are not sufficient on their own to address these concerns, and proposed the adoption of Data Portraits: records supporting membership inference on the data used to train a model. After discussing examples of existing solutions, we described our own time and space efficient version based on Bloom filters and host several demos. In future work, we hope to investigate applications of similar efficient data structures for indexing and counting. Our current implementation is minimal in that it aims to satisfy a form of membership testing and nothing more, such as full text indexing. We demonstrate its usefulness in case studies, and call for dataset and model creators to release some form of Data Portrait as part of their essential documentation.

## Acknowledgements

We thank the JHU HLTCOE for compute resources used in developing this work and the BigCode collaboration project for hosting the artifact documenting their Stack dataset. We thank students in the JHU CLSP for their feedback on this project, particularly Orion Weller, Nathaniel Weir, and Kate Sanders. Daniel Khashabi also provided valuable guidance in addressing reviewer feedback.

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

# A    Supplemental Documentation

This paper recommends best practices does not construct a new dataset or benchmark; thus not all parts of the recommended supplemental material (e.g. a model card or datasheet) are applicable. However, since we host an implementation and plan to distribute artifacts and code, we address similar relevant concerns:

- URL, landing page, and demo: dataportraits.org
- Code: to be released through the site above. Open source code will be hosted on GitHub.
- Hosting and Preservation: We will maintain the existing interactive web interfaces for at least one year from publication. Code will be available for long-term distribution through GitHub.

# B    Sketch Misses

In Figure 2, consider the query string: `defg`. This string lies on the boundary of the n-grams split during corpus processing. No stride 1 width 4 ngrams extracted from that query will match the database. However, if the query string were expanded to length $2 \cdot width - 1$, note that it would necessarily intersect at least one of the hashed n-grams. For example, `defghij` will match at `fghi`. In this way, given a long enough query string, our query protocol guarantees that at least one match will be found if the query string of interest does occur in the corpus.

# C    Adversarial Matches

We describe a protocol for *chaining* matches together. In Figure 2, the three matches of `bcde`, `fghi`, `jklm` occur separated by $width$ indexes. Therefore we infer the whole string (formed by concatenating the three n-grams) was present in the corpus. However, this might not be true. If an adversary knew the details of the sketch width (and initial offset into the sequence), they could construct a document that embeds n-grams in different locations, such that a query string would appear to be present according to our protocol. For example, the sketch in Figure 2 would falsely infer that the string `fghibcde` is present in the corpus, since it is composed of two chosen matches. This is very unlikely in practice, given appropriately chosen widths and sketch resolutions. This is essentially a permutation attack, and a similar approach could be used to fool a BM-25 index.

# D    WMT Documents

Table 5 lists the full test set, doc id, and approximate longest match results from Figure 3.

# E    Counting Expected Matches

Consider a string of interest $S$, with length $N$ that is embedded in a larger document $D$. Matching $S$ with a strided Bloom filter with width $w$ will yield a chain of something around $\frac{N}{w}$ tiles (substrings of length $w$). For example, take a sketch with width $w = 50$ and a string with $N = 150$. If the tiles in $D$ are perfectly aligned on the boundaries of $S$, the Bloom filter will find $150/50 = 3$ matches. Perfectly aligned means that string $S$ begins at an offset in $D$ that is a multiple of $w$ — so when breaking $D$ into non-overlapping chunks (tiles) of size $w$, the start of some tile is also the start of $S$.

This perfect alignment might happen by chance, but most likely there will be some parts of $S$ that hang over the tile boundaries, meaning only some inner part of $S$ will match the hashed tiles. In Figure 2 the query string $S = abcdefghijklmn$ is not perfectly aligned. *a* and *n* hang over and thus the only complete tiles are the inner *bcde*, *fghi*, *jklm* tiles match.

Given the width and length of a string, we can calculate the expected number of matches for any possible alignment of the string. Note that there are $w$ possible alignments and each is equally likely.

A string of length $N$ modulo the tile width $w$ can be written as $N = aw + b$ where $a$ is the number of full tiles and $b$ is the remainder. Consider alignments other than the perfect one boundary. We have

Table 5: Full WMT overlap information.

| Test Set | doc_id | Longest |
|----------|--------|---------|
| wmt13.en-fr.ref | lemondefr/2012/12/01/275696 | 700 |
| wmt16.en-fi.ref | kaleva.fi.29723 | 400 |
| wmt16.en-de.ref | tagesspiegel.de.65447 | 350 |
| wmt20.en-iu.ref | nunatsiaq-20190930 | 300 |
| wmt12.en-de.ref | noroeste/2011/11/15/78596.html | 250 |
| wmt16.en-de.ref | borkenerzeitung.de.56604 | 250 |
| wmt14.en-fr.ref | 4bb85eb6281e0b19986de1d4f867e3ff | 250 |
| wmt15.en-ru.ref | 893-kommersant | 200 |
| wmt18.en-fi.ref | karjalainen.fi.65284 | 200 |
| wmt18.en-ru.ref | kommersant.324314 | 200 |
| wmt14.en-fr.ref | cd085bbb218a7afc1255b2b60a06692a | 200 |
| wmt15.en-de.ref | 14428-abendzeitung-muenchen.de | 200 |
| wmt15.en-ru.ref | 115-aif | 200 |
| wmt16.en-ru.ref | lgng.30237 | 150 |
| wmt16.en-ro.ref | ziare.ro.17378 | 150 |
| wmt15.en-ru.ref | 1375-rg.ru | 150 |
| wmt14.en-fr.ref | 90c566f54bf1076e6f539875d45d673c | 150 |
| wmt17.en-ru.ref | izvestiya.51251 | 150 |
| wmt16.en-ro.ref | hotnews.ro.8884 | 150 |
| wmt14.en-fr.ref | 96e21a07ed57d79665a35a548ef7d841 | 150 |
| wmt16.en-de.ref | abendzeitung-nuernberg.de.12297 | 150 |
| wmt17.en-de.ref | dw.47065 | 150 |
| wmt18.en-de.ref | handelsblatt.com.180784 | 150 |
| wmt17.en-de.ref | frankfurter-rundschau.70094 | 150 |
| wmt13.en-fr.ref | cyberpresse/2012/12/01/1564248 | 100 |

$b + 1$ alignments that produce $a$ matching tiles. We will also have $w - b - 1$ alignments that produce $a - 1$ tiles. Summing and cancelling terms, we have $(b+1)a + (w-b-1)(a-1) = aw - w + b + 1$ possible matching tiles. Substituting the length of the string simplifies to $N - w + 1$ possible matches, and since each $w$ alignment is equally likely, the expected number of matches is $E(N, w) = \frac{N-w+1}{w}$. The 4 possible alignments with 11 possible matching strings in the Figure 2 example are:

```
[abcd, efgh, ijkl] (missing mn)
[bcde, fghi, jklm] (missing a, n)
[cdef, ghij, klmn] (missing ab)
[defg, hijk] (missing abc, lmn)
```

