# OpenReview forum: "Data Portraits: Recording Foundation Model Training Data"
_NeurIPS.cc/2023/Track/Datasets_and_Benchmarks — NeurIPS 2023 Datasets and Benchmarks Poster_

### Official Review · Reviewer_RQmh · 2023-07-21
**Simple and Effective Method Tackling a Critical Problem in Foundation Model Research**

**Rating:** 7
**Confidence:** 3
**Correctness:** Yes.
**Clarity:** Yes.

**Strengths:**

- Clear motivation for Data Portraits, especially pertinent with the current trend of ML research
- Very simple but effective method applying existing data sketching approaches to LLM datasets
- Method is low-latency, requires little storage, supports matching and indexing, etc.
- Application to pertinent LLM datasets, alongside live, usable demo

**Additional Feedback:**

How do you envision extending Data Portraits beyond a Bloom Filter approach?

**Documentation:**

Yes.

**Ethics:**

No.

**Limitations:**

Yes.

**Opportunities For Improvement:**

- Why only consider Bloom filters and not their more modern variants (e.g. Partitioned Learned Bloom Filter, Learned Bloom Filters, etc.)?


**Relation To Prior Work:**

Yes.

**Summary And Contributions:**

The authors introduce the notion of Data Portraits, a form of documentation artifacts based on membership inferences, as a means to record training data. They introduce a specific instance of Data Portraits via Bloom filters and use it to document The Pile and The Stack, yielding the ability to answer critical questions about test set leakage and model plagiarism.

---

> ### Author Response · Authors · 2023-08-23
>
> Thank you for the feedback. In particular, we appreciate that you acknowledge the two parts to our proposal: that we first introduce “a form of documentation artifacts” and then “introduce a specific instance of Data Portraits”. This is exactly our intent, to first call for community adoption of indexing/membership testing documentation artifacts, and then to propose a minimal and lightweight solution - though we are happy to see adoption of any other documentation practices as well!
>
> > Why only consider Bloom filters and not their more modern variants? …
>
> We did not explore modern variants because we intended to propose a simple and minimal method. Basic Bloom filters are already extremely efficient. However, we do think that future work might benefit from additional succinct data structures (see paragraph below and response to YRzG). Learned Bloom filters (e.g. [1] among others) introduce a learned model that attempts to capture whether a key is in a set. From [1]: “*For example, a Bloom filter that represents a set of malicious URLs can benefit from a learned model that can distinguish malicious URLs from benign URLs. This model can be trained on URL features such as length of hostname, counts of special characters…*”
>
> Given that LLM datasets intend to span many domains, it is not clear whether we could obtain a suitable pre-filtering model. This would also introduce large latency overhead - [1] appears to use the sklearn RandomForest classifier, while our current implementation uses the murmur hash function through redis. Murmur is intentionally designed to use very few cpu cycles per byte.
>
> > How do you envision extending Data Portraits beyond a Bloom Filter approach?
>
> As mentioned in our response to R.YRzG, we hope to build further documentation tools that incorporate fuzzy matching or other approximate data structures. We also found that our current version is limited by data copying and random memory access. In an ideal Bloom filter, the hash function will distribute bits uniformly through a bit array. However, such random memory access is extremely cache inefficient, potentially causing cache misses. One Bloom filter variant is the Block Bloom filter [2], which uses an initial hash to retrieve a basic filter that fits entirely in CPU cache. A second round of hashing queries only that Bloom filter, ensuring that only the first memory access can suffer a cache miss. This implementation is more complex and may incur additional space overhead, but could support even larger datasets (or less lossy filters) with friendlier memory access patterns.
>
> [1] Vaidya et al. Partitioned Learned Bloom Filter, https://arxiv.org/abs/2006.03176
>
> [2] Putze, Sanders & Singler, Cache-, Hash- and Space-Efficient Bloom Filters, https://www.cs.amherst.edu/~ccmcgeoch/cs34/papers/cacheefficientbloomfilters-jea.pdf

---

### Official Review · Reviewer_KAn6 · 2023-07-24
**Paper=good; at concise summarizations of larger datasets**

**Rating:** 7
**Confidence:** 4
**Correctness:** The technique is clearly explained an…

**Strengths:**

This paper has a clear value proposition: a concise summary of a dataset to make it easy to detect specific contents.

The paper is a widely applicable data-centric technology rather than yet-another-dataset, which is highly welcome. It provides a useful tool for summarizing the content of a dataset, which can help with various forms of data pollution: test leakage, pii, etc.

The paper is well structured: the problem, existing solutions, ideal solution qualities, author's solution, simple example, relevant full-scale examples, use cases, and limitations.


**Additional Feedback:**

Nice work.

**Clarity:**

The paper is very well written. It's an easy read, and the simple example is nicely chosen and illustrated.

**Documentation:**

The authors' micro-site is really clear and well executed. Well done!

**Ethics:**

This paper is ethically positive: if the authors' technique was more widely adopted it would help reduce some ethical concerns in the field around dataset content.

**Limitations:**

The authors clearly and sufficiently explain the limits of their technique: it trades flexibility for efficiency, and is lossy.

**Opportunities For Improvement:**

Generally, this is a focused, well-executed paper without major flaws. It is practical, useful, and reasonably general, though somewhat incremental.

**Relation To Prior Work:**

The paper has well structured related work section and explains it's connection to that work.

**Summary And Contributions:**

The paper describes using bloom filter hashing to produce concise (~3% of full size) summaries of large datasets. They authors apply their approach to highly relevant examples: ThePile and TheStack.
They also provide several practical applications of technology such as estimating leakage from translation test sets. As noted by the authors, their approach is limited to detecting specific content and is lossy, but is relatively inexpensive to compute, store, and use.

---

> ### Author Response · Authors · 2023-08-23
>
> Thank you for the feedback! We are glad that our work is recognized as “widely applicable data-centric technology” and not “yet-another-dataset” and welcome any further feedback on the demo site.
>
> While you didn’t raise any specific concerns, you may find our new experiments and analysis interesting. In particular, we further demonstrate the relative inexpensiveness of our approach while also showing that our minimal implementation of membership testing is still effective even with lossy hash-based storage (Section 5 in the revised submission).
>
> > if the authors' technique was more widely adopted it would help reduce some ethical concerns in the field around dataset content.
>
> We certainly agree and hope to drive further community adoption of dataset documentation tools, whether ours or others!

---

### Official Review · Reviewer_g9wo · 2023-07-25
**Good work and idea overall; would benefit from thorough and in-depth analysis of the proposed approach.**

**Rating:** 5
**Confidence:** 3

**Strengths:**

The proposed method is simple, efficient, and the artifacts can be safe to redistribute. The authors use the bloom filter to index the corpus, which is an appropriate data structure for the task of data sketching. In addition, the authors adopt the strided bloom filter for further reducing the storage as well develop other proper mechanisms in the query stage for merging consecutive identified n-grams. Because the indexing is only hashes rather than plain text, it can be relatively safely distributed.

This work falls in the line of work for more transparent access and assessment of recent large language models (training data); the method and demo can be very helpful for different types users (content creators, researchers, and content consumers, according to the author, line 27) for both inspecting the outputs generated by the models, as well as checking if the target data exists in the training corpus.


**Additional Feedback:**

NA

**Clarity:**

The paper states the motivation, method, and impact clearly. The case studies would benefit from some reorganization.

**Correctness:**

See the comments in Opportunities For Improvement.


**Documentation:**

NA

**Limitations:**

Yes, the authors addressed the limitations of the method (Section 5 and appendix C).

One potential negative social impact that the authors don’t mention is that with one such system released, one can modify the model generated outputs based on the results of the system to avoid being detected.


**Opportunities For Improvement:**

While the proposed method is helpful, I find it is hard to compare the method with other methods.

Perhaps one solution is that the authors can start with a small (synthetic) dataset, and use all the previously mentioned methods to perform the indexing and search. This simple experiment can help produce some tangible numbers like the compression ratio, search speed, etc across different methods on the same set of data.

In addition, it would be great for the authors to ablate the choices of the Data Portraits – e.g., what are the impacts of choosing different strides during indexing in terms of false positive rate as well as efficiency.


**Relation To Prior Work:**

This paper properly contrasts with previous work.


**Summary And Contributions:**

This paper proposes a method for data sketching that can be used to determine whether a string exists inside the training corpus of a large language model. The proposed method is based on the bloom filter which is easy to implement while being very efficient. In addition, the authors conduct case studies that apply the Data Portraits to existing datasets and showcase it can be used for cases like identifying overlapping between training and evaluation data and attributing generated code.

---

> ### Author Response · Authors · 2023-08-23
>
> Thanks for your feedback! We appreciate your acknowledgement of our efforts to produce an artifact that does not leak data.
>
> In response to your request for more comparisons and a synthetic dataset experiment, we direct you to the newly added Section 5. We focus on comparing to a full text search solution since we were able to obtain access to a comparable index through personal correspondence. However, we advise caution when directly comparing performance. Hardware differences such as index sharding or available RAM can create large differences in performance for search indexes. And of course, a full text index can provide far more information than our minimal Bloom filter approach.
>
> We created a synthetic classification dataset by collecting positive samples from the Pile and negative samples from recently published NYTimes articles. Our results show that while other dataset documentation methods are sufficient to create a perfect classifier, our method is far faster. With regards to compression ratio, we do not know the exact size of the Lucene index though the implementation is extremely similar to the Roots Search Tool of Piktus et al. 2023.
>
> With regards to ablations of the false positive rate, we now discuss this in lines 351-354 of the revised paper. Decreasing the FPR by an order of magnitude increases the size of the structure to 36GB.
>
> We also acknowledge your insightful comment concerning avoiding plagiarism detection with a tool like ours. This is an important concern and we have added it to our limitations section!

---

### Official Review · Reviewer_YRzG · 2023-07-27
**Review for Data Portraits**

**Rating:** 5
**Confidence:** 4
**Correctness:** Yes. But I didn't check thoroughly.
**Clarity:** Yes.

**Strengths:**

1.Test set leakage is a significant concern as it affects benchmarks used for evaluating the performance of language models. The role of Data Portraits here is crucial, as it can identify the leaked portions of the test set and measure the overlap between the training and the test sets.

2.The implementation showcases both time and space efficiency while being utilized as a plagiarism checking tool for live users.

**Additional Feedback:**

See above.

**Documentation:**

Yes. But I didn't check thoroughly.

**Opportunities For Improvement:**

1.The method lacks novelty. Using a Bloom filter to store the hash representation of n-grams is a well-known approach[1]. It would be beneficial to provide additional explanation as to why these established methods cannot be directly applied to the training set of the LLM, and why your tool is necessary.

2.Some related work aspect has been neglected. While the comparison with full-text search algorithms is comprehensive, it lacks comparison with plagiarism detection algorithms, such as Winnowing[2], MinHash[3].

3.Sensitive to the query string. To check if it's part of the training set, the query string needs to have long, exact matches. This suggests that a slight difference between the query string and the string in the training set would hinder the tool's ability to perform membership testing. This limitation may restrict its application.

Questions
1.Could you offer more justification for choosing a sketch width of 50? This parameter is critical as it impacts both efficiency and the plagiarism detection threshold.

2.What is the false negative rate, or "sketch miss," of your artifact in case study?

[1]Talbot D, Brants T. Randomized language models via perfect hash functions[C]//Proceedings of ACL-08: HLT. 2008: 505-513.

[2]Schleimer S, Wilkerson D S, Aiken A. Winnowing: local algorithms for document fingerprinting[C]//Proceedings of the 2003 ACM SIGMOD international conference on Management of data. 2003: 76-85.

[3]Broder A Z. On the resemblance and containment of documents[C]//Proceedings. Compression and Complexity of SEQUENCES 1997 (Cat. No. 97TB100171). IEEE, 1997: 21-29.

**Relation To Prior Work:**

Yes.

**Summary And Contributions:**

The paper presents a tool designed for constructing Data Portraits, purposed for the efficient inspection of training data to determine if a specific text was part of the training set. This tool utilizes a bloom filter to store the hashed representation of tiled n-gram derived from the training data. When querying, the tool extracts n-grams from the input query string and matches them with the Bloom filter. In principle, this allows LLM providers to release a Data Portrait without revealing their exact training data. Despite its inability to perform indexing or fuzzy searching, the tool excels in speed and storage capacity. The paper demonstrates the construction of a Data Portrait on The Pile dataset and evaluates the degree of overlap with various test datasets. Additionally, the authors construct a Data Portrait on a subset of The Stack dataset, serving as the foundation for a live plagiarism detection service.

---

> ### Author Response · Authors · 2023-08-23
>
> > The method lacks novelty. Using a Bloom filter to store the hash representation of n-grams is a well-known approach…
>
> No existing method or toolkit around Bloom filters is designed explicitly for dataset documentation or provides an interface like ours. Our use of striding, chaining, and analysis of membership testing on text datasets further distinguishes this tool. We also analyze other tools in detail, highlighting their properties and tradeoffs.
>
> We agree prior work has established Bloom filters and similar structures as useful NLP tools. We have revised our prior work section to include more references, including Talbot and Osborne. Most related work uses Bloom filters or other sketches to store features, rather than designing for documentation purposes.
>
> > lacks comparison with plagiarism detection algorithms, such as Winnowing[2], MinHash[3].
>
> MinHash is an interesting possible extension, perhaps enabling another type of portrait! In this work we focus on exact match, rather than the fuzzy near-matching enabled by MinHash and similar methods. We are aware that MinHash has been used for fuzzy deduplication in other LLM literature and have clarified these references in our related work. To our knowledge however, no MinHash-on-LLM dataset index has been persisted for later distribution to the LLM community. Thus we were not able to gather comparable statistics such as those in Table 2 (size of various tools on text datasets).
>
> That said, we can offer further analysis. Recent literature (Falcon, https://arxiv.org/pdf/2306.01116.pdf) describes the resources used to deduplicate their datasets with MinHash and similar techniques: “We usually run with 10,000-20,000 vCPUs in the cluster, enabling rapid parallel processing”. In a controlled study on a small sample of data, we find the following statistics:
>
> - Data Size: 61.4 MiB
> - MinHash Index size: 10.98 MiB (following parameters of https://huggingface.co/blog/dedup)
> - Our Bloom filter: 1.14 MiB
>
> As expected, measured space usage is higher than the Bloom filter. However, it is still quite small. We plan to investigate “fuzzy” Data Portraits, possibly enabled by MinHash or approximate counting structures, but believe this is best suited for future work.
>
> > Could you offer more justification for choosing a sketch width of 50?
>
> Thank you for the recommendation. We add additional support to Section 4.1, lines 214-223. 50-grams match prior literature on deduplication and dataset sanitization. GPT-2 and 3 use token n-grams that approximate 50-character grams. We also find that 50-character grams tend to be nearly unique in a sample of Pile data.
>
> > Sensitive to the query string. To check if it's part of the training set, the query string needs to have long, exact matches. This suggests that a slight difference between the query string and the string in the training set would hinder the tool's ability to perform membership testing. This limitation may restrict its application.
>
> Concerns over long exact matches are valid and closely related to question 2 (regarding sketch misses). This is described in the paper as the danger that a string of interest “falls between” adjacent n-grams (thus neither would match). As we describe, this can be alleviated by limiting queries to strings of interest that are at least of length $2n-1$. $n$ is the n-gram width of the sketch, which is a parameter set prior to construction.
>
> We confirm this threshold with analysis on real text. We randomly extract sentences from Pile documents and divide them into two classes: longer than the critical threshold and shorter. We use sentences as these are linguistically meaningful spans of text, which may or may not align to our choice of n-gram width. For sentences longer than the critical threshold, we find that at least one match is always detected, as expected by our protocol. For sentences shorter than the critical threshold, only 40% produce a match. Note that our web UI will warn when inputs are too short.

---

> > ### Comment · Reviewer_YRzG · 2023-08-30
> >
> > Thanks for the responses.
> > Considering the novelty, I will keep the score as 5. Considering it is a tool that may benefit the community, I would not complain if this paper gets accepted.

---

### Official Review · Reviewer_FHQe · 2023-07-27
**Data Portraits: Recording Foundation Model Training Data**

**Rating:** 5
**Confidence:** 4

**Strengths:**

- The general idea of the paper is interesting. The problem of publicly profiling the pretraining corpora is especially relevant in the context of the scaling pretraining data of large language models, and the data leakage explored by the community.

- The proposed solution is more optimal than other considered methods, and allows to document data in the form of the substring hashes, without actually revealing the entire corpus.

**Additional Feedback:**

N/A

**Clarity:**

In general, the paper is well-written, although the discussion of the limitations and impact is limited (which is also understandable given the space constraints).

**Correctness:**

As mentioned above, the analysis could be extended to get a more complete picture of the work.

**Documentation:**

The current submission does not provide the codebase to support reproducibility of the experiments.

**Ethics:**

The intended use of the work is clear. As mentioned above, however, the paper does not discuss any *negative* societal impact.

**Limitations:**

- The paper does not discuss any potential negative societal impact.

- I believe that one of the limitations is the false positive rates (discussed in Section 3.1, but not in Section 5). What are the authors' thoughts on conducting an empirical experiment to calculate the false positive rate on at least one of the considered corpora?

**Opportunities For Improvement:**

1. While I understand that the goal of the experiments is to show that some test set examples might appear in the pretraining corpus (as also stated in L242--244), I believe the analysis might somehow be extended: e.g., analysing the metrics on the samples identified as leaked, or comparing the amount samples identified as leaked by other related tools.

2. Experiments in Section 4.3 might be consistent: e.g., analysing the overlap statistics as in Section 4.4 (Tab. 3)

3. The web-interface is a great contribution; however, it is not clear whether the proposed method is only available as the interface, or also as an easy-to-use library? The current submission also does not provide any codebase for reproducing the experiments.

4. The resource is publicly available since March 6, but the license is not yet stated.

**Minor**
- misspellings (e.g., probabalistic in Table 1)
- formatting links/urls

**Relation To Prior Work:**

The paper discusses the background and related work in Section 2, and provides comparison of Data Portraits and other tools in Section 4. It is understandable how the proposed solution differs from the other tools from the technical perspective.

**Summary And Contributions:**

This paper introduces **Data Portraits**, a web-interface for documenting pretraining corpora of large language models, which are generally not publicly available.

The main contributions are:
1) the development of the toolkit based on the Bloom filters, along with the web-interface which supports analysis of whether a given sample (e.g., a text or a code snippet) appears in the pretraining corpus;
2) the authors compare their tool with other publicly available tools, and conduct a few experiments on identifying the amount of overlap between test sets from downstream datasets and the Pile and the Stack corpora.

The results indicate that the proposed solution is more optimal than existing related data documentation tools, and there is a significant overlap between the WMT and XSum test sets and the Pile corpus.

**Data Portraits** is available at: https://dataportraits.org.

---

> ### Author Response · Authors · 2023-08-23
>
> Based on your request for extended analysis, comparisons to other tools, and “an empirical experiment to calculate the false positive rate” we have completed a set of new experiments (Section 5 of the revised paper).
>
> We construct a classification task and dataset using text sampled from the Pile and text certainly not present in the Pile. We build a classification rule based on the longest common substring, applied to both our method and a full text index provided through personal correspondence. This allows for an experiment “comparing the amount samples identified as leaked by other related tools” as requested. We hope this suits your concerns and we welcome other suggested experiments!
>
> > I believe that one of the limitations is the false positive rates (discussed in Section 3.1, but not in Section 5). What are the authors' thoughts on conducting an empirical experiment to calculate the false positive rate on at least one of the considered corpora?
>
> Thank you for noticing that omission from our limitations section, it has now been added! See the following paragraph for further commentary on negative impacts. We also conduct a small false positive study using the Lucene index we recently obtained access to (detailed in the revised Section 5.1). We query the Bloom filter with novel NYTimes text and record all n-grams flagged by the sketch. The false positive rate is 0.0007, slightly less than the theoretical rate of 0.001. We also briefly discuss space usage for other theoretical false positive rates.
>
> > The paper does not discuss any potential negative societal impact.
>
> We add a note about negative impacts to Limitation (Section 6). A system such as ours could allow malicious users to modify text to avoid plagiarism detection systems.
>
> To clarify our intended release: We commit to hosting the web interface using our own computational resources for at least one year following publication of this paper. This will provide membership testing for various important pretraining corpora, even for lay-users. We intended to publicly release binary files containing the Bloom filter. This is a standard serialized datatype from Redis (https://redis.io/docs/data-types/probabilistic/bloom-filter/) which means it can be queried directly or with our Portraits interface code. We are also exploring API support (similar to the current demo site) and believe it is possible to host a limited research API endpoint using our current resources as well.

---

> > ### Comment · Reviewer_FHQe · 2023-08-29
> > **Response: revision**
> >
> > Thank you for addressing some of my suggestions. I confirm that I have read the revision.
> >
> > Could you please share your thoughts/suggestions/answers to the questions that are not addressed in your responses and revision? In particular:
> >
> > - The web-interface is a great contribution; however, it is not clear whether the proposed method is only available as the interface, or also as an easy-to-use library? The current submission also does not provide any codebase for reproducing the experiments.
> > - The current submission does not provide the codebase to support reproducibility of the experiments.
> > - analysing the metrics on the samples identified as leaked
> > - Experiments in Section 4.3 might be consistent: e.g., analysing the overlap statistics as in Section 4.4 (Tab. 3)
> >
> > Thanks.

---

> > > ### Author Response · Authors · 2023-08-29
> > > **Primary contribution is not a website**
> > >
> > > Responding to: "a web-interface for documenting pretraining corpora of large language models, which are generally not publicly available"
> > >
> > > We want to stress that the primary contribution of this article is conceptual, aligning with existing community standards like Model Cards and Data Sheets. That we provide a particular implementation is meant only to illustrate the viability of this proposal. By no means is this meant to be the only potential implementation of a Data Portrait, and in fact we enumerate multiple potential solutions (e.g., "grep", Elastic Search, ...). We are arguing that LLM creators -- in particular the subset responsible for data curation -- should ensure the capability for post-hoc example search.
> > >
> > > When Meta, Anthropic, ..., release new models, we believe they should release a Portrait alongside it. This is the all-caps point we would like to make at NeurIPS to conference goers.

---

> > > ### Author Response · Authors · 2023-08-30
> > >
> > > We are glad that you appreciate the web interface. We also hope the new results address your concerns as you asked that "the analysis might somehow be extended", "...or comparing the amount samples identified as leaked by other related tools", and for "the authors' thoughts on conducting an empirical experiment to calculate the false positive rate". For your additional points (1-4):
> > >
> > > **1,2:** We describe our planned release in the final paragraph of our response. This release is complicated by two factors: the size of the data structure(~60 GB for both datasets) and packaging issues related to the dependency on a database system with compiled extensions (redis). We don't have an easy way to distribute the binary file at the moment, though we are exploring solutions and intend to do so.
> > >
> > > **That said, we have uploaded code providing a succinct API [here](https://dataportraits.org/portraits-neurips/index.html).** To address point 4 and point 2 about reproducibility, we include a log of generating similar results to Table 3 on HumanEval (see following, `datasketch.py` in the code). This code relies on loading the Pile and Stack binary bloom filters, which we cannot easily include here due to their size.
> > >
> > > Please consider this code upload private for the duration of review, to be replaced with a fully engineered package upon later full release. The fully engineered package will contain at least the functionality in the above release. The primary API interface is shown here:
> > >
> > > ```
> > > sketch = RedisBFSketch('stack.march-no-pii.tight-2.code.50-50.bf')
> > > reports = sketch.contains_from_text(documents)
> > > ```
> > > Where `reports` then contains membership information for all n-grams extracted from a list of strings `documents`.
> > >
> > > **3.** We are not sure what you meant by "analysing the metrics on the samples". If you mean compare metrics with similar methods, we believe the additional experiments in Section 5 cover this.
> > >
> > > **If you mean measure model performance on examples with high overlap, we believe this is best suited for later work.** This work calls for better documentation tools and provides one possible implementation. We hope that this enables others (including model trainers) to study performance. A study of performance involves many variables: appropriate dataset (generation, classification, code generation), models (large model with accompanying open dataset), and analysis metrics (accuracy? calibration?).
> > >
> > > **4.** To address your point about the consistency between 4.3 and 4.4, they are meant to illustrate *different* aspects of "what could you do with a rapid membership testing tool?". **However, using the code from 1. we generate similar metrics as Table 3 but using both the Stack system and the Pile system on the HumanEval test set for code.** Results are included in the prior zip, as are examples of long overlaps. We find similar amounts of span overlap (stack: 10%, pile: 9%, latency: ~250ms). While there are many spans that overlap, instance level inspection reveals that these are primarily "generic" snippets of code.
> > >
> > > Here is one such example (prime number generation) in our web interface: [primes](https://stack.dataportraits.org/#JTIwJTIwJTIwJTIwcHJpbWVzJTIwJTNEJTIwJTVCJTVEJTBBJTIwJTIwJTIwJTIwZm9yJTIwaSUyMGluJTIwcmFuZ2UoMiUyQyUyMG4pJTNBJTBBJTIwJTIwJTIwJTIwJTIwJTIwJTIwJTIwaXNfcHJpbWUlMjAlM0QlMjBUcnVlJTBBJTIwJTIwJTIwJTIwJTIwJTIwJTIwJTIwZm9yJTIwaiUyMGluJTIwcmFuZ2UoMiUyQyUyMGkpJTNBJTBBJTIwJTIwJTIwJTIwJTIwJTIwJTIwJTIwJTIwJTIwJTIwJTIwaWYlMjBpJTIwJTI1JTIwaiUyMCUzRCUzRCUyMDAlM0ElMEElMjAlMjAlMjAlMjAlMjAlMjAlMjAlMjAlMjAlMjAlMjAlMjAlMjAlMjAlMjAlMjBpc19wcmltZSUyMCUzRCUyMEZhbHNlJTBBJTIwJTIwJTIwJTIwJTIwJTIwJTIwJTIwJTIwJTIwJTIwJTIwJTIwJTIwJTIwJTIwYnJlYWslMEElMjAlMjAlMjAlMjAlMjAlMjAlMjAlMjBpZiUyMGlzX3ByaW1lJTNBJTBBJTIwJTIwJTIwJTIwJTIwJTIwJTIwJTIwJTIwJTIwJTIwJTIwcHJpbWVzLmFwcGVuZChpKSUwQSUyMCUyMCUyMCUyMHJldHVybiUyMHByaW1lcw==)
> > >
> > > While 4.4 and 4.3 are meant to illustrate distinct uses, we would consider reorganizing Section 4 if results such as the above are of substantial interest to reviewers.

---

> > > > ### Comment · Reviewer_FHQe · 2023-08-30
> > > > **Response 2: revision**
> > > >
> > > > Thank you for your answers.
> > > >
> > > > **1, 2**. I understand the limitations due to computational costs, etc. My initial intention about the codebase release was to highlight that it would be helpful if your method would be publicly available. While it is debatable on how to organize the storing/distribution of the corpora given the computational and space limitations etc., I think these problems can be addressed by the community as well. For example, the model developers could release their pretraining corpus using their own budget/corpus storage & distribution resources **and your codebase**, and these corpora could be somehow downloadable via your web-interface?
> > > >
> > > > It was (and it is) not clear to me how the community could use your approach from the technical point of view, and what are the recommendations for the intended use. There is also no documentation of the code you have attached.
> > > >
> > > > I guess it might be helpful to describe the limitations associated with the usage of Data Portraits in the paper, as they are currently unresolved.
> > > >
> > > > **3**. Yes, I meant analyzing the model performance on examples with a high overlap.  I believe it might as well illustrate distinct uses, as you mention in your response, which has a direct application of your method.
> > > >
> > > > **4**. Thank you, it is clear to me now that you want to show different aspects of the method application. I think it would be nice to explicitly describe it in the corresponding sections.

---

### Author Response · Authors · 2023-08-23
**New paper section with additional analysis, further measurements and statistics in individual responses.**

We thank the reviewers for their feedback and appreciate that all reviewers recognize our aims to provide a discussion of documentation tools and a practical method for increasing LLM dataset transparency. We also appreciate that reviewers acknowledged our key factors and design choices: efficiency, redistribution of material, and our demo site.

Reviewers FHQe, YRzG, g9wo all requested additional analysis around comparisons to other tools, false positive rates, parameter choices, and requested additional general evidence for the efficacy of our proposed approach. We emphasize that our contributions are two-fold: we call for membership testing as a documentation best practice and provide a lightweight and minimal implementation. While we review other tools in detail, these are often designed for different purposes and thus difficult to compare empirically. That said, we introduce an additional section of the paper (Section 5, Analysis) where we:

- Create a membership testing binary classification dataset
- Use our method and a full text search method to create classifiers, assessing their performance and efficiency
- Measure and analyze observed false positive rates.

We summarize this section below and provide further answers in individual responses:

To create a classification dataset, we need a source of high quality text that is certainly not in the Pile. We manually gather paragraphs of text from New York Times articles published in the past 2 weeks. We next build classifiers using the Bloom filter and a full text search engine comparable to some of the discussed related work. We find that both classifiers can achieve perfect classification on our (admittedly synthetic) task, *but that the Bloom filter approach is approximately 700 times faster.* Perhaps we could construct a more difficult classification task (e.g. with ambiguous text or quotations), but the runtime differences would remain. We find the false positive rate on a per-ngram basis is 0.0007 and that there are no false positive chains (longer sequences of consecutive matching n-grams).

Other additional analysis and the corresponding response:
- Impact of false positive rate on structure size. (R.g9wo, also of interest to R.FHQe)
- Justification of the selected n-gram width (R.YRzG, also of interest to R.g9wo)
- Statistics of a possible MinHash solution (R.YRzG)
- Impact of input/query length as sentences (R.YRzG)

Other paper changes:
- Note usage of MinHash and other data sketches in prior work
- Note potential malicious uses of our work (avoiding plagiarism detection)
- Minor spelling and formatting corrections.

We appreciate the positive feedback on the demonstration site and encourage the reviewers to explore that interface if they have not already. Links to pre-filled examples are present in the paper and project landing page at https://dataportraits.org/

---

> ### Author Response · Authors · 2023-08-28
> **Any additional comments?**
>
> We thank the reviewers for their comments so far. Since the response period is soon coming to a close, please let us know if you have any further comments on our revised submission and additional experiments.

---

### Decision · Program_Chairs · 2023-09-22

**Decision:**

Accept (Poster)

**Comment:**

This is an interesting proposal that has potentially significant ramifications. The authors call for the inclusion of data portraits as a general practice when companies produce "foundation" or base models, as a transparency mechanism that can help external entities understand the inclusion of specific training samples in the model. This of course also has significant implications on data provenance questions, and is also likely useful for other considerations, such as when model trainers want to reduce the inclusion of AI generated data in their models, for fear of model error application or model collapse.

The reviewers achieve rough consensus that bloom filters applied to this context is an interesting approach. The question of why not more modern variants is quite valid. And while the authors response of prioritizing simplicity is sufficient, it would be interesting to at least see a bit of discussion on potential benefits from different bloom filter variants (there have been so many), and if there are interesting new directions for followup work in those directions.

The authors made significant edits to the paper in the rebuttal process, and the paper has been improved.

Overall, this is a well written paper with strong potential for real impact on model transparency in this timely topic.